# Reconstructing 6-hourly PM$_{2.5}$ datasets from 1960 to 2020 in China

Junting Zhong[1], Xiaoye Zhang[1, 4*], Ke Gui[1], Jie Liao[2], Ye Fei[2], Lipeng Jiang[3], Lifeng Guo[1], Liangke Liu[5], Huizheng Che[1], Yaqiang Wang[1], Deying Wang[1], Zijiang Zhou[2]

[1]State Key Laboratory of Severe Weather & Key Laboratory of Atmospheric Chemistry of CMA, Chinese Academy of Meteorological Sciences, Beijing, 100081, China

[2] National Meteorological Information Center, Beijing, 100081, China

[3] Earth System Numerical Prediction Center, Beijing, 100081, China

[4]Center for Excellence in Regional Atmospheric Environment, IUE, Chinese Academy of Sciences, Xiamen, 361021, China.

[5]Department of Earth System Science, Tsinghua University, Beijing 100084, China

Correspondence to: Xiaoye Zhang (xiaoye@cma.gov.cn)

## Abstract

Fine particulate matter ($PM_{2.5}$) has altered radiation balance on earth and raised environmental and health risks for decades but has only been monitored widely since 2013 in China. Historical long-term $PM_{2.5}$ records with high temporal resolution are essential but lacking for both research and environmental management. Here, we reconstruct a site-based $PM_{2.5}$ dataset at 6-hour intervals from 1960 to 2020 that combines long-term visibility, conventional meteorological observations, emissions, and elevation. The $PM_{2.5}$ concentration at each site is estimated based on an advanced machine learning model, LightGBM, that takes advantage of spatial features from 20 surrounding meteorological stations. Our model's performance is comparable to or even better than those of previous studies in by-year cross validation (CV) ($R^2=0.7$) and spatial CV ($R^2=0.76$) and is more advantageous in long-term records and high temporal resolution. This model also reconstructs a $0.25°×0.25°$, 6-hourly, gridded $PM_{2.5}$ dataset by incorporating spatial features. The results show $PM_{2.5}$ pollution worsens gradually or maintains before 2010 from an interdecadal scale but mitigates in the following decade. Although the turning points vary in different regions, $PM_{2.5}$ mass concentrations in key regions decreased significantly after 2013 due to clean air actions. In particular, the annual average value of $PM_{2.5}$ in 2020 is nearly the lowest since 1960. These two $PM_{2.5}$ datasets (publicly available at https://doi.org/10.5281/zenodo.6372847) provide spatiotemporal variations at high resolution, which lay the foundation for research studies associated with air pollution, climate change, and atmospheric chemical reanalysis.

# 1  Introduction

In the past decades, anthropogenic emissions of reactive gases and aerosols have been emitted increasingly in the atmosphere and thus led to a substantial increase in fine particulate matter ($PM_{2.5}$). Increased $PM_{2.5}$ has strongly interacted with solar radiation through absorption and scattering, thereby reducing visibility and influencing the earth's radiance balance. Inhalable $PM_{2.5}$ has increased human morbidity and mortality through penetrating the respiratory system (Pope et al., 2002; Beelen et al., 2007; Chen et al., 2016b). To evaluate the impacts of $PM_{2.5}$ pollution on environment, climate, and health, the primary concern is to understand the spatiotemporal variations of $PM_{2.5}$ concentrations. Namely, extended $PM_{2.5}$ records with high temporal resolution lay the foundation for research studies associated with air pollution, climate change, and environmental health. Nevertheless, it was not until 2013 that the Ministry of Ecology and Environment (MEE) established a nationwide $PM_{2.5}$ monitoring network. Long-term, accurate historical $PM_{2.5}$ datasets are lacking for both research and environmental management.

Chemical transport models (CTMs) are expected to simulate the spatial and temporal variations of $PM_{2.5}$ with reasonable emission inventories inputted. However, significant uncertainties still exist in historical emission inventories and physicochemical mechanisms, which resulted in inevitable biases in the simulated absolute values of $PM_{2.5}$. Satellite-based aerosol optical depth (AOD), which measures the aerosol extinction of the solar beam, is an indicator of ground-level aerosols. AOD data products from Moderate Resolution Imaging Spectroradiometer (MODIS) have broad spatial coverage and relatively long observation periods (~ 20 years). Therefore, assimilating satellite-retrieved AOD to construct atmospheric chemical reanalysis is a practical approach to reducing $PM_{2.5}$ biases. In recent years, several international aerosol reanalysis datasets have been developed preliminarily, including the reanalysis data produced by the Copernicus Atmosphere Monitoring Service (CAMS) from the European Centre for Medium-Range Weather Forecasts (ECMWF) (Inness et al., 2019), the Modern-Era Retrospective analysis for Research and Applications, Version 2 (MERRA-2) from the National Aeronautics and Space Administration (NASA) (Gelaro et al., 2017; Randles et al., 2017), aerosol reanalysis from the Navy Aerosol Analysis and Prediction System (NAAPS) (Lynch et al., 2016) and the Japanese Reanalysis for Aerosol (JRAero) from the Japanese Meteorological Agency (Yumimoto et al., 2017). In particular, CAMS produced gridded $PM_1$, $PM_{2.5}$, and $PM_{10}$ data at 80 km resolution since 2003 by assimilating satellite retrievals of total AOD, total tropospheric $NO_2$ column, total $O_3$ column, CO column, and vertical profiles (Inness et al., 2019). MERRA-2 reanalysis includes $PM_{2.5}$ and $PM_{10}$ at 50 km resolution since 1980 by assimilating ground-based and satellite-retrieval (Gelaro et al., 2017; Randles et al., 2017). NAAPS generates gridded AOD data at ~100 km resolution from 2003 to 2013 by assimilating satellite-based AOD products (Lynch et al., 2016). JRAero provides $PM_{2.5}$ and $PM_{10}$ at ~100 km resolution from 2011 to 2015 by assimilating satellite AOD data (Yumimoto et al., 2017). These reanalysis data have contributed significantly to research in aerosol-related fields. However, there are still some weaknesses in accuracy, spatial resolution, time span, and types of assimilated data. In China, the highest horizontal resolution of the four reanalysis is only 50 km, and this coarse grid setting may not be sufficient to capture the spatial differences in atmospheric pollutants at regional scales. In terms of the type of aerosol data assimilation, these reanalysis data mainly assimilate satellite-based and ground-based AOD, and do not take into account ground $PM_{2.5}$ observations.

To overcome the reanalysis's weaknesses in low spatial resolution and high biases, numerical

researchers focus on constructing relatively long-term PM$_{2.5}$ datasets based on machine learning techniques that fuse multisource data, including satellite-retrieved AOD, CTM simulations, and even atmospheric chemical reanalysis. For example, Ma et al. (2016) estimated daily PM$_{2.5}$ records at 0.1° resolution between 2004-2013 with MODIS AOD. Liang et al. (2020) rebuilt monthly PM$_{2.5}$ concentrations at 1 km resolution during 2000-2016 based on the multiangle implementation of atmospheric correction (MAIAC) from MODIS and reanalysis AOD and PM$_{2.5}$ data from MERRA-2. Geng et al. (2021) reconstructed daily, 10 km PM$_{2.5}$ data between 2000-2020 with MODIS AOD and CTM simulations. Wei et al. (2021a) regenerated monthly, 1 km PM$_{2.5}$ records between 2000-2018 based on MAIAC AOD. Huang et al. (2021) estimated 1 km × 1 km PM$_{2.5}$ concentrations daily between 2013-2019 based on MAIAC AOD and CTM outputs. However, some inherent limitations in satellited-based AOD are challenging to overcome. Due to the low sampling frequency of satellite-retrieved AOD, AOD-based PM$_{2.5}$ datasets are limited to a maximum temporal resolution of one day. With AOD over land unavailable before 2000, these PM$_{2.5}$ datasets can only be back-calculated to 2000 at the earliest. Although recent studies focus on estimating hourly PM$_{2.5}$ during the daytime based on AOD from geostationary satellites like Himawari 8 (Chen et al., 2019; Yan et al., 2020; Wang et al., 2021; Wei et al., 2021b), obtained PM$_{2.5}$ datasets can only extend for several years, and the data is missing at night or with cloud cover.

Compared with satellite data, ground-based meteorological observations have the advantages of long sequence time, high temporal resolution, and good data integrity. In China, the national meteorological observation network of the China Meteorological Administration (CMA) was established in the 1950s and is capable of continuously observing 6-hourly meteorological data on visibility and conventional meteorological variables, including temperature, pressure, wind, and relative humidity (RH). The number of national stations exceeded 2,000 in 1960 and stabilized at around 2,450 afterward. Studies have shown that visibility and conventional meteorological variables are closely related to PM$_{2.5}$ (Zhang et al., 2013a; Zhang et al., 2013b; Zhang et al., 2015; Wang et al., 2018; Zhu et al., 2018; Zhong et al., 2018). For example, low wind speed is highly unfavorable to the horizontal diffusion of pollutants (Zhang et al., 2013b). The increase in RH favors the hygroscopic growth of PM$_{2.5}$ and also promotes the accelerated conversion of gaseous precursors to particulate matter, leading to a rapid increase in PM$_{2.5}$ concentrations (Pilinis et al., 1989; Ervens et al., 2011; Kuang et al., 2016). Atmospheric visibility is directly related to PM$_{2.5}$ mass concentrations under dry conditions and non-linearly related to PM$_{2.5}$ and RH under humid conditions (Wang et al., 2019). Therefore, better results may be achieved if these ground-based meteorological data can be used to estimate historical PM$_{2.5}$ data in China. Liu et al. (2017) first estimated monthly visibility-based PM$_{2.5}$ concentrations between 1957-1964 and 1973-2014 based on 674 publicly available meteorological stations. Gui et al. (2020) constructed a virtual daily PM$_{2.5}$ network at 1180 meteorological sites between 2017-2018. Our previous research also shows that the visibility-based machine learning model that takes advantage of spatial features has great potential in reconstructing historical PM$_{2.5}$ datasets with long-term records and high temporal resolution (Zhong et al., 2021). In this study, we reconstruct a site-based PM$_{2.5}$ dataset at 6-hour intervals from 1960 to 2020 based on long-term visibility and conventional meteorological observations from ~2450 national stations, together with emissions and elevation. The PM$_{2.5}$ concentration at each site is estimated based on a Light Gradient Boosting Machine (LightGBM) model that takes advantage of spatial features from 20 surrounding meteorological stations. By incorporating spatial features, this model also reconstructs a 0.25°×0.25°, 6-hourly, gridded PM$_{2.5}$ dataset. These two PM$_{2.5}$

123    datasets provide spatiotemporal variations at high resolution, which constitute the basis for research

124    studies associated with air pollution, climate change, and atmospheric chemical reanalysis.

125

## 2 Data and Methods

### 2.1 Multisource input data

*Observational PM$_{2.5}$ data*. The MEE began laying out a PM$_{2.5}$ monitoring network in January 2013, expanding the scope from key regions including the North China Plain (NCP), the Yangtze River Delta (YRD), the Pearl River Delta (PRD), and the Sichuan Basin (SB) as well as municipalities directly under the Central Government and provincial capitals, to 113 key and model cities for environmental protection, and eventually to all cities above prefecture level, with the number of observation sites expanded from the initial 520 to over 1,600. Since then, PM$_{2.5}$ mass concentrations have been recorded continuously using the β-absorption methods or a micro-oscillating balance following a standard protocol (Huang et al., 2021). Hourly PM$_{2.5}$ data of all sites between 2013-2020 are collected from the China National Environmental Monitoring Center (CNEMC, http://www.cnemc.cn). To produce high-quality PM$_{2.5}$ data, a series of quality controls were conducted, including integrity checking, duplicate rejection, and outlier handling. All sites with a proportion of valid PM$_{2.5}$ records exceeding 60% were considered. For each site, identical data for 3 consecutive hours were excluded first, and PM$_{2.5}$ values over three standard deviations from 24-hour and 3-day moving average were regarded as outliers and discarded then. Eventually, PM$_{2.5}$ data from 1485 sites remained for model development and application. In addition, pre-2013 PM$_{2.5}$ measurements in US embassies in Beijing and Shanghai are used for independent validation evaluations (http://www.stateair.net/web/historical).

*Visibility and conventional meteorological data*. The CMA established a national meteorological observation network in the 1950s, with the station number exceeding 2000 at the beginning and stabilizing at ~2,450 afterward. The observation network can continuously record meteorological data on visibility and conventional meteorological variables, including temperature, pressure, wind, and RH. In recent years, meteorological observations, including 6-hourly records between 1960-2020 and gradually increasing hourly records after 2013, have been collected from the National Meteorological Information Center (NMIC). Due to the inconsistency of visibility data in terms of observation methods, we conducted a series of data conversions to ensure continuous and consistent data. Visibility data recorded on a scale ranging from 0 to 9 between 1960-1979 were converted to numerical data based on probability density distributions. Specifically, the probability density distribution of visibility for each of the ten years before and after 1980 was calculated at first. The numerical visibility from 1980 to 1989 was graded into classes, with the median value of each class being the corresponding value for each station, and finally, the class observations were converted into numerical observations. From September 2013 to 2016, visibility measurements gradually shifted from 6-hourly manual observations to 1-hourly automatic observations site-by-site. In keeping with manual measurements, the automatic records, which are slighter lower than manual measurements, were calibrated by dividing 0.75 following the guideline from the CMA (Cma, 2014).

*Emission inventories and elevation*. Historical anthropogenic emissions from 1960-2012 are taken from Peking global emission inventories, developed using a bottom-up approach with spatial resolution at 0.1°×0.1° and temporal resolution at 1-month intervals (http://inventory.pku.edu.cn) (Chen et al., 2016a; Huang et al., 2014; Huang et al., 2015; Wang et al., 2014). Current

anthropogenic emissions during 2013-2020 are from the multiresolution Emission Inventory in China (MEIC, http://meicmodel.org) (Zhang et al., 2009; Zheng et al., 2018; Zheng et al., 2021). Six emission variables from these two inventories are used as inputs for model development, including $PM_{2.5}$, NOx, $SO_2$, $NH_3$, BC, OC, and CO. Thirty-meter elevation data are collected from the Global Digital Elevation Model (GDEM) version 2 (https://earthexplorer.usgs.gov). Both emission and elevation data are interpolated from grids to sites to match existing $PM_{2.5}$ sites.

*Auxiliary data*. Monthly Normalized Difference Vegetation Index (NDVI) products are downloaded from Level-1 and Atmosphere Archive & Distribution System Distributed Active Archive Center (LADDS DAAC, https://ladsweb.modaps.eosdis.nasa.gov). Land cover classification data are taken from National Geographic Information Resources Catalogue Service System (https://www.webmap.cn/mapDataAction.do?method=globalLandCover). Population data are taken from the Gridded Population of the World version 4 (GPWv4, https://sedac.ciesin.columbia.edu/data/collection/gpw-v4) and are calibrated based on the total population in China City Yearbooks. NDVI, Land cover, and population data are also interpolated according to $PM_{2.5}$ sites and trained as inputs for model development. However, during the model training process, we found that these data had little or no improvement in the hindcast capability of the model, and the time span of these data is insufficient for long-term historical retrieval. Hence, these auxiliary data are not used in model building.

## 2.2 Spatiotemporal feature extraction

For each $PM_{2.5}$ site, we extract five variables as temporal inputs, including year, month, day, hour, and day of year. The longitude and latitude variables are taken out as location inputs (Fig. 1b). Visibility, RH, and temperature from the nearest meteorological station of each $PM_{2.5}$ are used as basic meteorological inputs. The distance between these two sites was also added as a feature. In addition to the influence of the nearest meteorological station, $PM_{2.5}$ concentrations at a site are also affected by surrounding conditions. For example, transport of pollution due to air movement is the main cause of heavy pollution episodes in the early stage (Zhong et al., 2017; Zhong et al., 2018). Hence, we need to consider spatial effects from surrounding meteorological stations. Our previous study developed a novel feature engineering approach, which incorporated surrounding impact by extracting spatial features (Zhong et al., 2021). Specifically, the remaining 19 nearest stations were matched for each $PM_{2.5}$ site, except the nearest meteorological station. Five variables, including longitude, latitude, temperature, visibility, and RH, were selected from the 19 stations. Then, we calculated the maximum, the minimum, the average value, the skewness value, and the standard deviation for each of the five variables. These produced features, which take advantage of surrounding conditions, are also considered as inputs. After spatiotemporal feature extraction, a total of 71 features were used as inputs for model training. To reduce computation and training time with guaranteed accuracy, the top 40 features in order of importance during small-sample-testing processes are used for the following model training and hindcasting. These features included visibility, temporal features, spatial features, emission features, and elevation.

## 2.3 Gridded input construction

In the previous construction of input features for $PM_{2.5}$ sites, we used location information,

time information, meteorological information from 20 surrounding meteorological stations,
emission information, and elevation. If we assume that each cell in grid cells is a virtual $PM_{2.5}$ site,
then it is possible to generate input features for each grid point. After the model is trained based on
input features and $PM_{2.5}$ concentrations at real $PM_{2.5}$ sites, we can feed the gridded input data into
the model in turn and consequently construct a gridded $PM_{2.5}$ network. Therefore, we define a grid
area at $0.25°×0.25°$ with longitude from $70°$ E to $150°$ E and latitude from $10°$ N to $60°$ N and
select the grid points covering mainland China. For each grid point, we performed spatiotemporal
feature extraction and generated the same 71 input features as those of real $PM_{2.5}$ sites.

## 2.4 Model description

LightGBM is one of the state-of-the-art gradient boosting frameworks with better accuracy,
lower memory usage, faster training speed, and capability of handling large-scale data (Ke et al.,
2017). Our previous research used this machine learning model to predict $PM_{2.5}$ mass concentrations,
which shows an unprecedented predictive capacity on hourly, daily, monthly, and annual
timescales(Zhong et al., 2021). This study will continue to use this algorithm and previously tuned
hyperparameters for model development (Zhong et al., 2021). For hindcasting historical $PM_{2.5}$
datasets prior to 2013, a LightGBM model is trained and validated based on $PM_{2.5}$ observations and
feature inputs from 2013 to 2020. The hindcast capability is validated using cross-validation
methods, which are standard methods for parameter tuning and model validation in machine
learning. The training dataset is divided into several parts, one of them is used as test data, and the
remaining parts are used as training data in turn. Each result yields a corresponding evaluation value,
which is then averaged to provide an estimate of the model's accuracy. This estimation is quantified
by two metrics: the coefficient of determination ($R^2$) and root-mean-square error (RMSE). The
hindcast capability is also validated using $PM_{2.5}$ observations from the US embassies in Beijing and
Shanghai, which have been observing $PM_{2.5}$ data since as early as 2008. After model training and
validation, historical 6-hourly input data are inputted into this model to reconstruct a site-based
$PM_{2.5}$ dataset at 6-hour intervals from 1960 to 2020; and gridded input data are inputted into the
model to reconstruct a $0.25°×0.25°$, 6-hourly, gridded $PM_{2.5}$ dataset. The daily, monthly, yearly, and
decadal average $PM_{2.5}$ concentrations for each site and each grid are also calculated based on the
two datasets. Monthly-average values were obtained with daily values no less than 20 days;
otherwise, they will be missing. Year-average values were calculated with 12 valid month values,
and decadal-average values were calculated with 10 valid year-average values. The flowchart for
reconstructing $PM_{2.5}$ datasets is shown in Fig. 1.

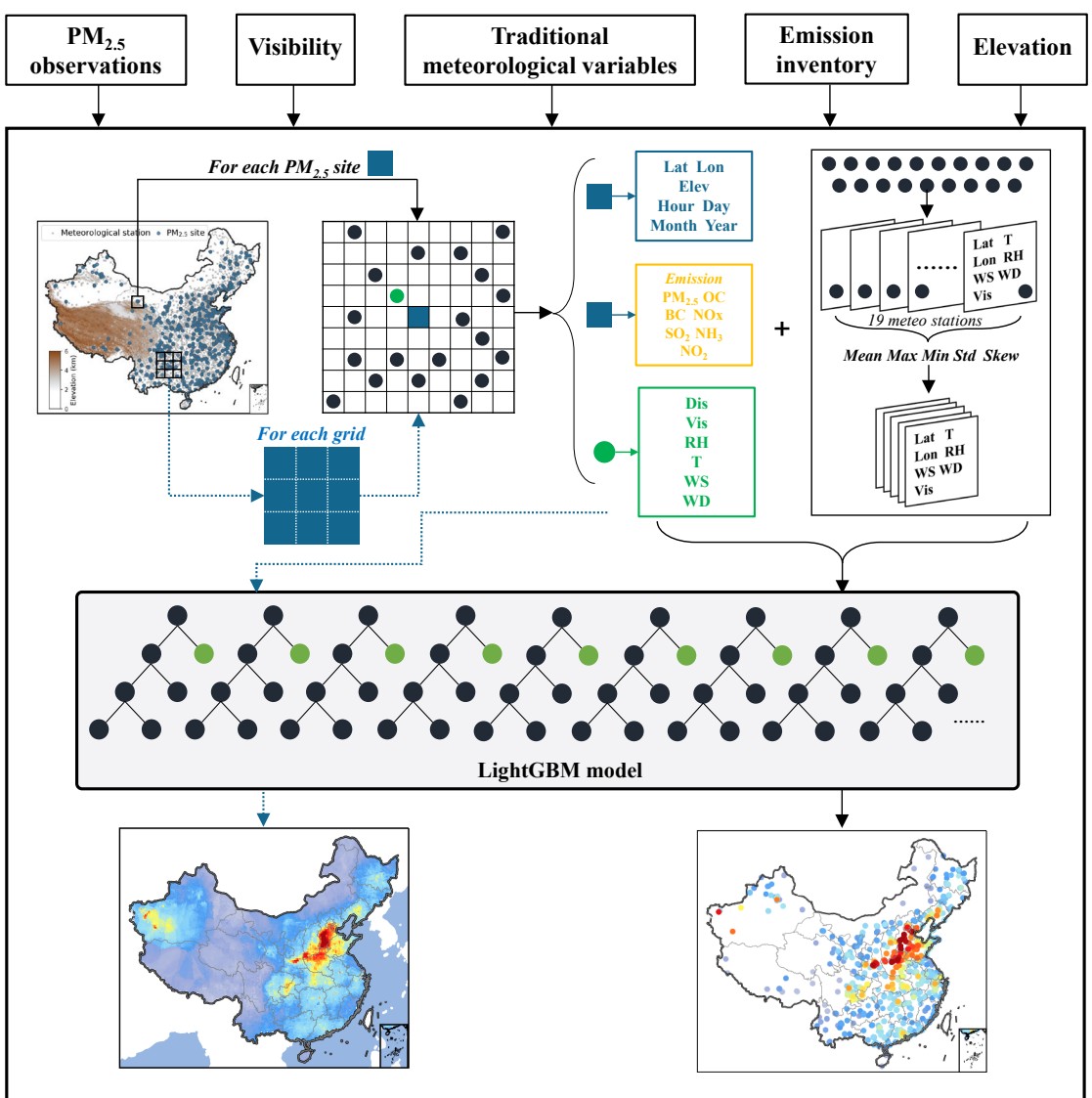

**Fig. 1** A conceptual scheme for constructing long-term historical site-based and gridded PM₂.₅ records based on long-term visibility, conventional meteorological observations, emissions, and elevation.

# 3 Results and Discussion

## 3.1 Evaluation of model hindcast performance

The hindcast performance of our model is evaluated using two CV methods, including 10-fold CV and by-year CV. The 10-fold CV partitions the original training datasets into 10 subsamples, one of which is retained as the validation data in turn for testing the model, and the remaining 9 subsamples are used as training data. This method is the most common CV that can be compared with results in other studies. However, 10-fold CV often overestimates the model's ability to hindcast continuous historical data. Therefore, we also use by-year CV, during which one year of data is selected sequentially for testing, and the remaining data are used for model training. This method is specifically designed to evaluate the hindcast capability of the model.

**Tab. 1** Model performance in primary predictors, temporal resolution, and hindcast capability compared with other national $PM_{2.5}$ datasets in China.

| Related studies | Primary predictors | Temporal resolution | CV type | CV resolution | CV $R^2$ | CV RMSE |
|---|---|---|---|---|---|---|
| Ma et al., 2016 | AOD | daily (2004-2013) | 10-fold CV | daily | 0.79 | 27.40 |
| | | | by-year CV | | 0.41 | |
| Fang et al., 2016 | AOD | daily (2013-2014) | 10-fold CV | daily | 0.80 | 22.80 |
| Liu et al., 2017 | Visibility | monthly (1957-1964, 1973-2014) | 10-fold CV | monthly | 0.71 | 25.62 |
| Xiao et al., 2018 | AOD | daily (2013-2017) | 10-fold CV | daily | 0.79 | 21.00 |
| Xue et al., 2019 | AOD、CTM outputs | daily (2000-2016) | by-year CV | daily | 0.61 | 27.80 |
| Liang et al., 2020 | AOD | monthly (2000-2016) | 10-fold CV | monthly | 0.93 | 6.20 |
| Huang et al., 2021 | AOD、CTM outputs | daily (2013-2019) | 10-fold CV | daily | 0.87-0.88 | 11.90-21.90 |
| | | | by-year CV | | 0.62 | 27.70 |
| Wei et al, 2021 | AOD | monthly (2000-2020) | 10-fold CV | monthly | 0.86–0.90 | 10.00-18.40 |
| | | | by-year CV | | 0.80 | 11.26 |
| van Donkelaar et al. 2021 | AOD、CTM outputs | monthly (1998-2020) | Non-CV | yearly | 0.69 | 11.90 |
| Geng et al., 2021 | AOD、CTM outputs | daily (2000-2020) | out-of-bag CV | daily | 0.80-0.88 | 13.90-22.10 |
| | | | by-year CV | | 0.58 | 27.50 |
| Bai et al., 2022 | AOD | daily (2000-2020) | 10-fold CV | daily | 0.79 | 20.04 |
| Our study | Visibility | 6-hourly (1960-2020) | 10-fold CV | hourly/6-hourly | 0.79 | 20.07 |
| | | | | 6-hourly | 0.78 | 21.14 |
| | | | | daily | 0.85 | 16.11 |
| | | | | monthly | 0.92 | 7.90 |
| | | | by-year CV | hourly/6-hourly | 0.70 | 26.36 |
| | | | | 6-hourly | 0.71 | 25.63 |
| | | | | daily | 0.78 | 20.90 |
| | | | | monthly | 0.83 | 13.37 |

Table 1 compares our dataset and the available datasets in primary predictors, temporal resolution, and CV results(Ma et al., 2016; Fang et al., 2016; Liu et al., 2017; Xiao et al., 2018; Xue et al., 2019; Liang et al., 2020; Huang et al., 2021; Wei et al., 2021a; Van Donkelaar et al., 2021; Geng et al., 2021; Bai et al., 2022). AOD-based datasets are only available from around 2000 at the earliest, with temporal resolutions ranging from daily scale to monthly scale. In contrast, our visibility-based dataset spans 61 years from 1960 to 2020 at 6-hourly intervals, showing a clear advantage in terms of time span and resolution. The $R^2$ and RMSE values of our 10-fold CV results are 0.78 and 21.14 μg m$^{-3}$ for 6-hourly estimations, respectively, which indicates our model is quite robust in estimating $PM_{2.5}$. Due to a reduction in data amount, the $R^2$ and RMSE values further improved to 0.85 and 16.11 μg m$^{-3}$ for daily estimations and 0.92 and 7.90 μg m$^{-3}$ for monthly estimations. This result is comparable to or even better than those of other available datasets whose 10-fold CV $R^2$ ranges from 0.61 to 0.80 on a daily scale and from 0.71 to 0.93 on a monthly scale. Our by-year CV's $R^2$ and RMSE values are 0.71 and 25.63 μg m$^{-3}$ for 6-hourly estimations, which indicates our model is still robust in hindcast performance. The by-year CV $R^2$ values for daily and monthly estimations (0.78 and 0.83) are higher than those in other available datasets (0.41-0.62 and

0.80), which might be partly attributed to spatial feature extraction and the large volume of our
training dataset. Zhong et al. (2021) has shown that extracting spatial features can result in a better
hindcast performance by fully representing dimensional heterogeneity. Compared to hundreds of
thousands to millions of training samples in AOD-based models, the training samples for the
visibility-based model are over 100 million. An increase in the order of magnitude for training
datasets will yield better results in machine learning.

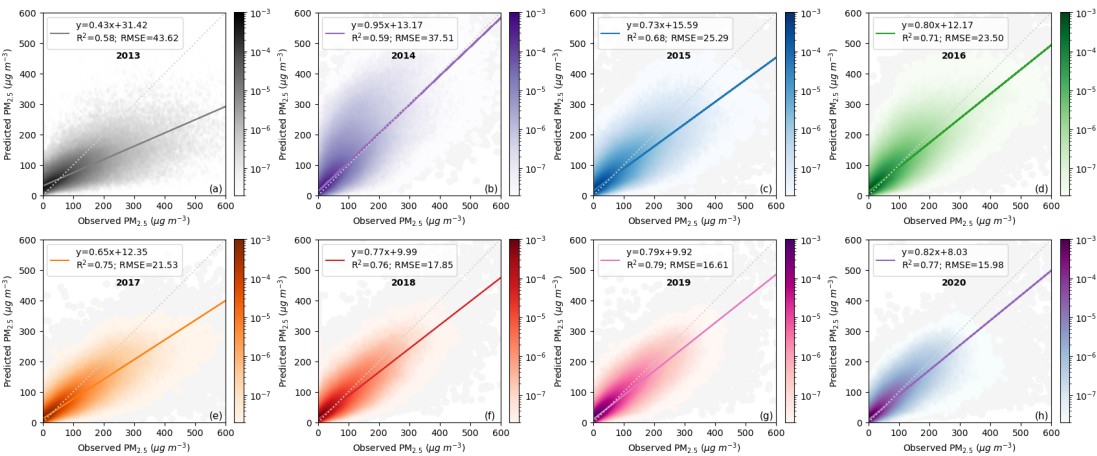

**Fig. 2** Density scatterplots of observed $PM_{2.5}$ and estimated $PM_{2.5}$ across China for by-year CV
from 2013 to 2020. The time resolution for CV results is hourly and 6-hourly between 2013-2017
and hourly between 2017-2020. (Colors are probability distribution densities).
The refined by-year CV results for each year between 2013-2020 are shown in Fig. 2. The
by-year CV $R^2$ lies between 0.58 and 0.79, with better hindcast performance after 2014. The
potential reasons why the $R^2$ value in 2013 is slightly lower than those in other years are as follows.
First, the $PM_{2.5}$ observation network was just established in 2013, during which dehumidification
systems, processing procedures, and data quality control methods are incomplete, and therefore the
overall data quality cannot be guaranteed. With the improvement of the observation network after
2014, both the quality and quantity of observations increase significantly. This situation where data
quality is relatively low initially but increases over time is also found in $O_3$ observations. Second,
the CMA began to convert some of the manual visibility observations to automatic observations in
2013, during which there were also some irregular procedures in instrument equipment, observation
steps, and data quality control. Lastly, although we have corrected the biases between manual and
automated observations, some biases may still exist. However, the biases are further reduced as we
integrate all manual visibility observations in 2013 into our training dataset.
The model's hindcast capability is further evaluated independently using pre-2013 $PM_{2.5}$
observations. For the $PM_{2.5}$ data currently available, only the US embassies in Beijing and Shanghai
have at least one year's $PM_{2.5}$ observations. Therefore, $PM_{2.5}$ data from these two sites are applied
as an independent evaluation dataset. Figure 3 shows our estimated $PM_{2.5}$ are in close agreement
with in-situ measurements in Beijing and Shanghai, where the overall $R^2$ between observations and
estimations is 0.74 and 0.79, respectively. For each year between 2008-2012 in Beijing, the $R^2$
values fluctuated between 0.70 and 0.81, reflecting a stable and accurate by-year hindcast capability.
As shown in Fig. 3 (c-h), the low values, high values, and temporal variations in $PM_{2.5}$
measurements are all well estimated. In particular, $PM_{2.5}$ measurements are lacking at the US

Embassy in Beijing in early 2008 and around 2009, but our model can provide reasonable and continuous estimations to fill in the gaps. This ability can also be used to fill in missing $PM_{2.5}$ observations of MEE from 2013 onwards, building a complete $PM_{2.5}$ dataset. Overall, the independent validation results show that historical $PM_{2.5}$ data can be well reconstructed by our model.

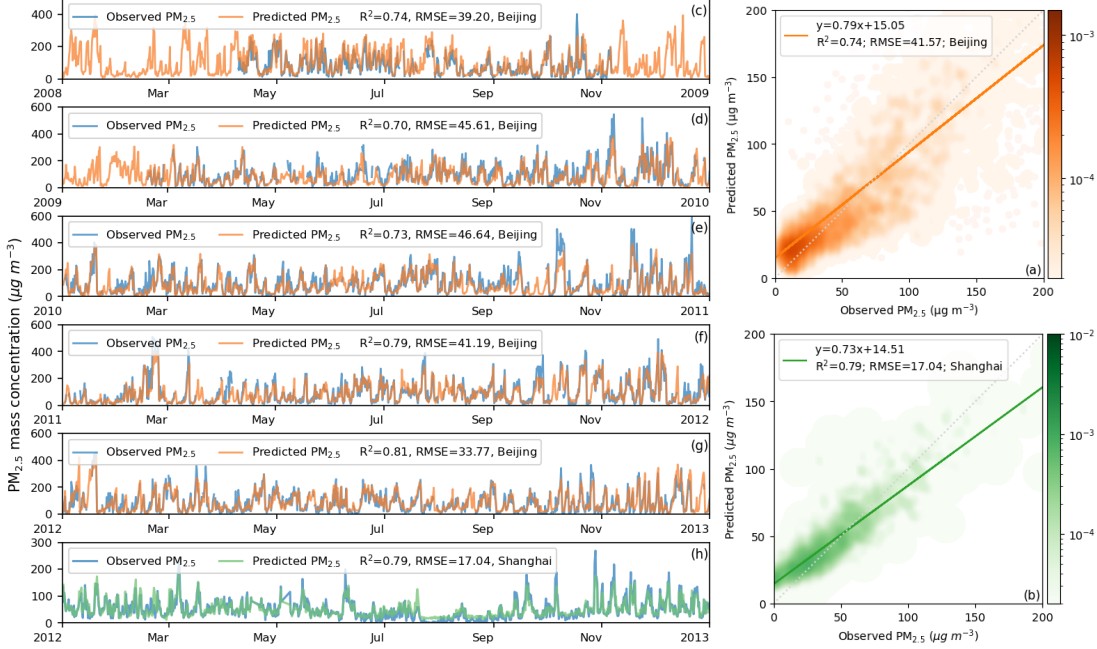

**Fig. 3** (a) Density scatterplots of observed $PM_{2.5}$ and estimated $PM_{2.5}$ between 2008-2012 at the US Embassy in Beijing; (b) Density scatterplots of observed $PM_{2.5}$ and estimated $PM_{2.5}$ in 2012 at the US Embassy in Shanghai; (c-g) Timeseries of observed $PM_{2.5}$ and estimated $PM_{2.5}$ for each year between 2008-2012 at the US Embassy in Beijing; and (h) Timeseries of observed $PM_{2.5}$ and estimated $PM_{2.5}$ for each year in 2012 at the US Embassy in Shanghai.

The model's ability to make $PM_{2.5}$ predictions at locations outside the scope of the training stations is evaluated by spatial CV. For spatial CV, all the monitoring stations are randomly divided into five subsets, and the model is trained using data from four subsets and tested on the data from the remaining subset each time. As shown in Fig. 4, the $R^2$ for spatial cross-validation in different groups is between 0.75 and 0.79, reflecting robust predictive power for $PM_{2.5}$ concentrations at sites outside the training sites. Our previous study also examined this predictive ability using $PM_{2.5}$ data from 23 untouched regional $PM_{2.5}$ stations (Zhong et al., 2021).

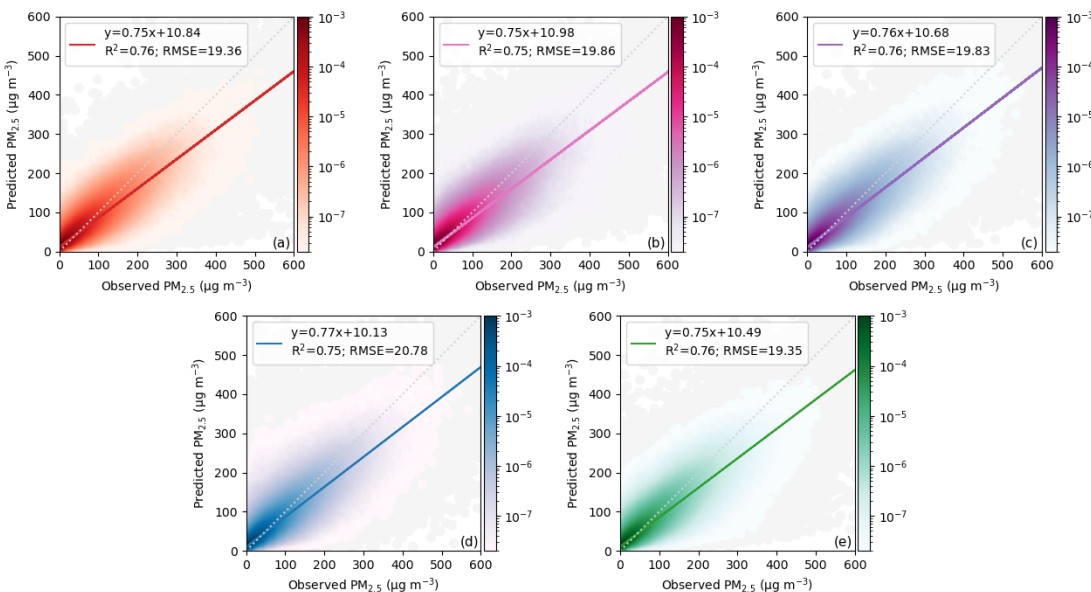

**Fig. 4** Density scatterplots of observed PM$_{2.5}$ and estimated PM$_{2.5}$ for each group of spatial CV results

## 3.2 Spatiotemporal variations in the site-based PM$_{2.5}$ dataset during 1960-2020

Figure 5 shows the spatiotemporal variations in annual average site-based PM$_{2.5}$ between 1960-2020. The trend of PM$_{2.5}$ in China experiences three major stages, corresponding to a slow increase under low concentrations between 1960-1978, a continuous accumulation with high concentrations reached between 1979-2013, and a rapid decrease between 2014-2020. During the first stage, though PM$_{2.5}$ pollution occurred in parts of the NCP and the Guanzhong Plain (GZP), PM$_{2.5}$ concentrations remain low in the vast majority of areas. This is mainly because anthropogenic emissions of PM$_{2.5}$ precursors and primary PM$_{2.5}$ grow slowly at a low base, resulting in relatively low total emissions in different regions. However, PM$_{2.5}$ pollution still occurring in the NCP and GZP, even with relatively low emissions, indicates the low environmental capacity of these two regions. During the second stage, PM$_{2.5}$ reached an unprecedentedly high concentration after a continuous increase in nearly all regions in China. The heaviest PM$_{2.5}$ pollution occurred in the NCP and the GZP. The SB and the Northeast China Plain (NeCP) are the polluted regions with the next highest PM$_{2.5}$ pollution. Even the YRD and the PRD also experienced PM$_{2.5}$ pollution during this stage. This worsening of PM$_{2.5}$ pollution is closely associated with massive anthropogenic emissions from rapidly increasing living and industrial activities after reform and opening-up policies. From 1979 to 2013, primary PM$_{2.5}$, NOx, SO$_2$, NH$_3$, BC, OC, and CO from the Peking emission inventory increased by 98%, 457%, 159%, 117%, 45%, -22%, and 243%, respectively. Despite a slow reduction in SO$_2$ after 2006, the total anthropogenic emissions each year still increased and thereby caused high-level PM$_{2.5}$ pollution after 2006. The results indicate that air pollutants cannot be emitted without restraint, even in regions with high atmospheric capacity. Otherwise, PM$_{2.5}$ pollution will inevitably occur. In addition to anthropogenic emissions, sand and dust storms, resulting in high PM$_{2.5}$ concentrations in western Xinjiang, worsened PM$_{2.5}$ pollution by trans-regional transport from the desert regions. During the last stage, PM$_{2.5}$ decreased nationwide with the mass concentrations in nearly all stations approximately or below 35 ug m$^{-3}$ in 2020, even in the NCP and the GZP with limited environmental

capacity. The substantial declines in PM$_{2.5}$ illustrate the effectiveness of implementing the toughest-ever clean air policy in China. The spatiotemporal variations of PM$_{2.5}$ between 1960-2020 clearly show the long-term impact of economic development and energy consumption on our air quality and the effectiveness of recent years' unprecedented emission control policies.

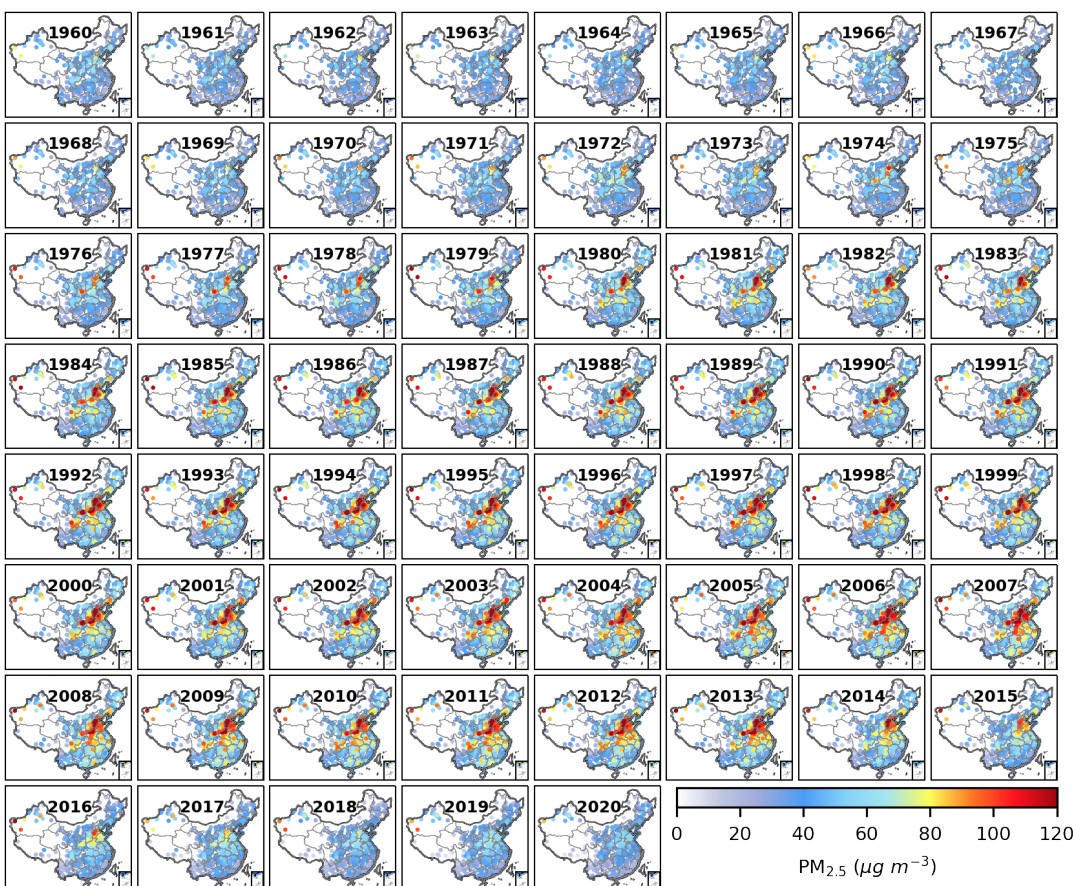

**Fig. 5** Spatial distribution of annual average PM$_{2.5}$ mass concentration at 1485 stations from 1960 to 2020

The specific turning points in annual PM$_{2.5}$ concentrations for different regions were investigated additionally. Figure 6 shows the temporal variations in national-average monthly and yearly PM$_{2.5}$ mass concentrations and regional average 6-hourly, monthly, and yearly PM$_{2.5}$ mass concentrations in "2+26" cities of the NCP, the YRD, the PRD, and the SB. The national-average yearly PM$_{2.5}$ reached a peak of 67 ug m$^{-3}$ in 2007, declined in 2008, and then remained steady until 2013. A sharp fall followed after 2014, with PM$_{2.5}$ concentrations decreasing from 63 ug m$^{-3}$ in 2013 to 34 ug m$^{-3}$ in 2020. The annual PM$_{2.5}$ concentrations in the "2+26" cities also experienced similar changes with a peak in 2007 and a reduction in 2008, which might be related to emission reduction for the Beijing Olympics in 2008. For the YRD, the maximum value of PM$_{2.5}$ mass concentration occurred in 2013 without a striking peak in 2007. For the PRD, the annual PM$_{2.5}$ concentrations increased steadily between 1960-1978, then rose more and more steeply in the following years with a steep increase in 2003 and 2004 and peaked in 2004. A steady decrease with slight fluctuation occurred from 2005 to 2013, and then a sharp fall followed after 2014. This trend is different from that in the "2+26" cities and the YRD. For the SB, the turning point occurred in 2013, before which

374 the annual PM$_{2.5}$ concentrations increased steadily and remained steady.

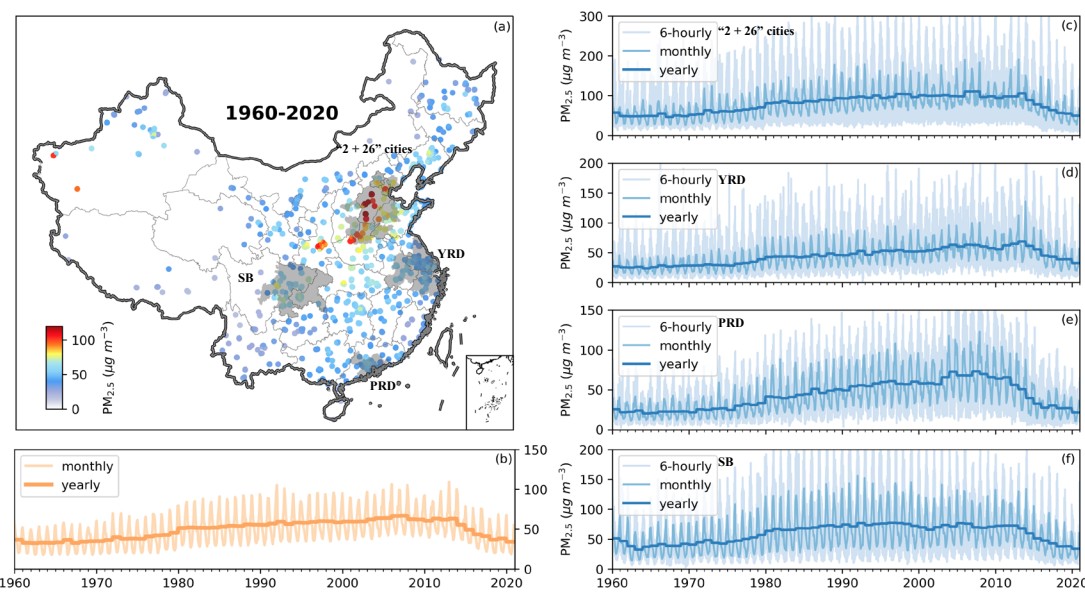

**Fig. 6** (a) Spatial distribution of average PM$_{2.5}$ mass concentrations between 1960-2020; (b-f)
Timeseries of average PM$_{2.5}$ mass concentrations for all sites in China (b), "2+26" cities (c),
Yangtze River Delta (d), Pearl River Delta (e) and Sichuan Basin (f), respectively.

## 3.3 Detailed spatial distributions from gridded PM$_{2.5}$ datasets

Figure 7 shows the annual spatial variations in 0.25°×0.25° gridded PM$_{2.5}$ between 1960-2020.
Compared to site-based distributions, gridded PM$_{2.5}$ can portray the spatiotemporal variations in a
clearer and more detailed way. For example, the most widespread and heaviest PM$_{2.5}$ pollution in
western Xinjiang occurred in 1979. This abnormal pollution corresponds to the historical
construction of northern severe dust storms, which recorded the event with the largest affected areas
in April 1979 (Zhou and Zhang, 2003). As exposed to nearly the most frequent air stagnation in
winter due to terrain and meteorological conditions (Wang et al., 2018), the NCP is the region with
PM$_{2.5}$ pollution first to appear and last to disappear except areas affected by dust storms (Fig. 7).
For year-to-year comparisons, it can be clearly seen that PM$_{2.5}$ concentrations in the NCP decreased
slightly from 2007 to 2008 and from 2012 to 2013, respectively, and decreased significantly in 2014
relative to 2013. The PM$_{2.5}$ reduction is insignificant from 2015 to 2016 but striking from 2016 to
2017. In 2020, the nationwide PM$_{2.5}$ concentrations are comparable to those in 1960s and close to
the lowest level ever recorded in almost 61 years.

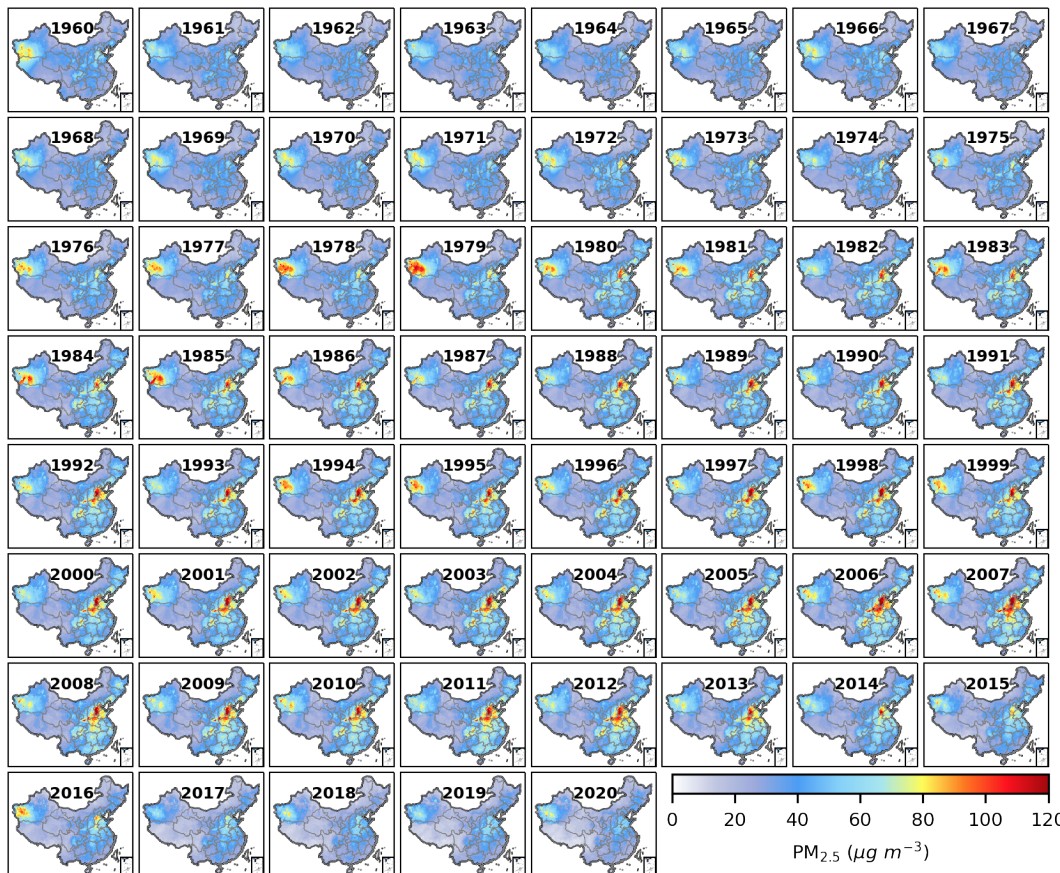

**Fig. 7** Gridded distribution of annual average PM$_{2.5}$ mass concentration from 1960 to 2020

Figure 8 shows inter-decadal spatial variations in gridded PM$_{2.5}$ between 1961-2020. PM$_{2.5}$ concentrations maintained at low levels in most areas over the first decade and increased to a certain extent in the NCP and western Xinjiang over the second decade. In the following decades, PM$_{2.5}$ pollution has worsened significantly in several key regions, including the NCP, the GZP, and the SB. This worsening was maintained until the last decade, during which PM$_{2.5}$ pollution mitigates significantly in nearly all populous and polluted regions in eastern China.

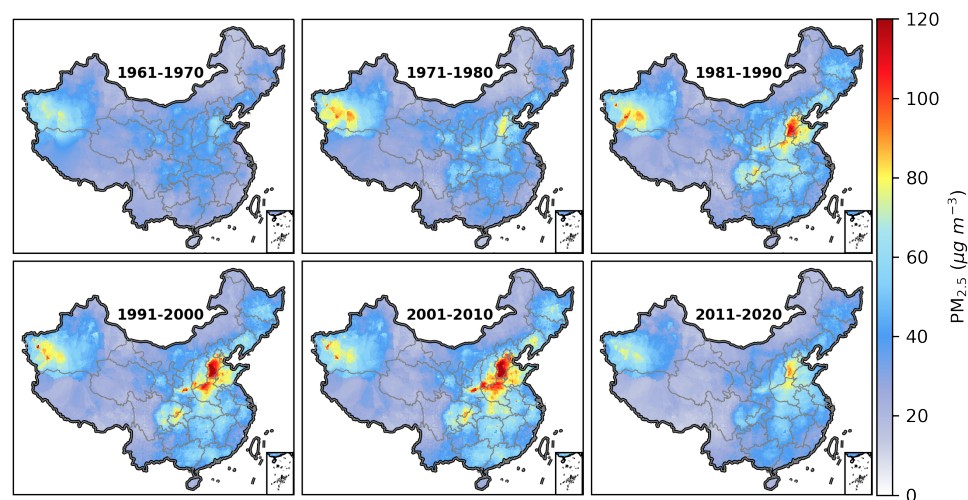

**Fig. 8** Gridded distribution of decadal average PM$_{2.5}$ mass concentration from 1960 to 2020

The multi-year trend of our gridded $PM_{2.5}$ dataset is also compared with those of publicly available datasets, including the TAP data (Geng et al., 2021), the GEFPM data (Van Donkelaar et al., 2021), the LGHAP data (Bai et al., 2022), and the CHAP data (Wei et al., 2021a), which have been interpolated to the same grid resolution. Figure 9 shows the spatial distributions of $PM_{2.5}$ from those datasets at 5-year intervals between 2000-2020. One consistent trend across all datasets was that nationwide $PM_{2.5}$ mass concentrations experienced an increase following a decrease from 2000 to 2020. However, the turning points are different for different datasets. From 2010 to 2015, $PM_{2.5}$ pollution alleviated for TAP, CHAP, and our data but worsened for GEFPM and LGHAP. For the time (2015 and 2020) with ground observations available, all $PM_{2.5}$ data show similar spatial distributions with the most severe pollution in the NCP in 2015 and significant improvement in nationwide air pollution in 2020. For the years (2000, 2005, and 2010) when ground observations were unavailable, significant disparities in pollution levels and regional distribution emerged from different datasets. Specifically, the LGHAP data are significantly lower than other data, while the TAP data are higher than others in nearly all regions except western Xinjiang. In western Xinjiang, $PM_{2.5}$ concentrations from the GEFPM data are the highest among all the datasets. Due to a lack of ground $PM_{2.5}$ observations before 2000, it is challenging to determine which dataset has the least bias and more reasonable distributions. In the future, applying ensemble average to multi-datasets might be an effective way to eliminate systematic bias.

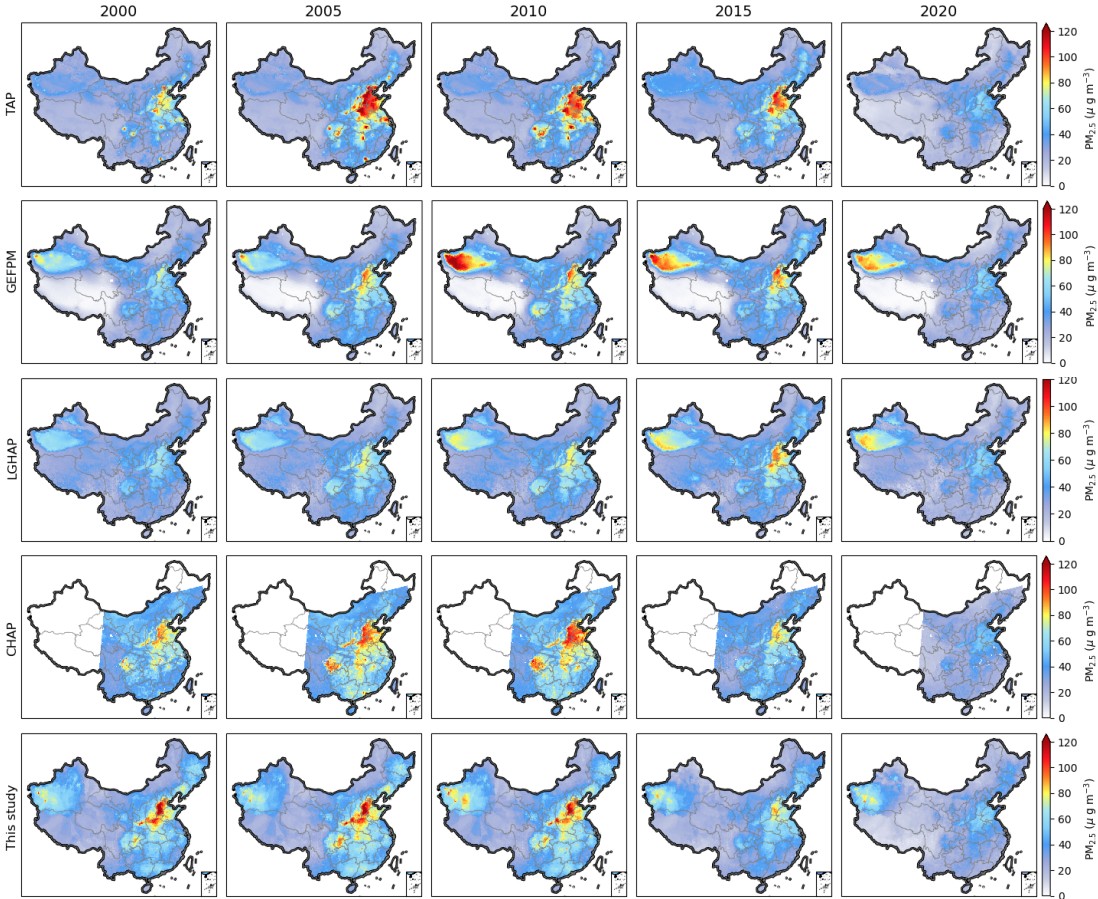

**Fig. 9** Distribution of reconstructed $PM_{2.5}$ by different $PM_{2.5}$ datasets in 2000, 2005, 2010, 2015, and 2020. From top to down are TAP, GEFPM, LGHAP, CHAP, and our dataset.

## 4 Data availability

The 6-hourly PM$_{2.5}$ datasets from 1960 to 2020, including site-based and gridded data, are publicly accessible. Daily, monthly, and yearly sited-based and gridded PM$_{2.5}$ datasets are also provided. The sited-based PM$_{2.5}$ dataset is in the CSV format, and the gridded dataset PM$_{2.5}$ is in the NETCDF format. All of them are available at https://doi.org/10.5281/zenodo.6372847 (Zhong et al., 2022).

## 5 Conclusion

This study is among the first to generate long-term site-based and gridded PM$_{2.5}$ datasets between 1960-2020 with 6-hourly resolution, based on long-term visibility, conventional meteorological observations, emissions, and elevation. A new feature engineering method that takes advantage of spatial features from 20 surrounding meteorological stations is employed in our LightGBM model to incorporate spatial effects of meteorological conditions. For by-year CV, the R$^2$ values of our model are 0.71, 0.78, and 0.83 for 6-hourly, daily, and monthly estimations, respectively, which are higher than those in other available datasets (0.41-0.62). This hindcast capability is further evaluated independently using pre-2013 PM$_{2.5}$ data of 6 years from US embassies in Beijing and Shanghai. The low values, high values, and temporal variations in US-embassy PM$_{2.5}$ measurements are all well estimated with the overall R$^2$ being 0.74 and 0.79 in Beijing and Shanghai, respectively. Both by-year CV and independent validation show that our model has a stable by-year hindcast capability and can reconstruct historical PM$_{2.5}$ data in a relatively accurate way. Our datasets show that PM$_{2.5}$ variations in China experience a slow increase under low concentrations between 1960-1978, a continuous accumulation with high concentrations reached between 1979-2013, and a rapid decrease between 2014-2020. The worsening of PM$_{2.5}$ pollution is closely associated with massive anthropogenic emissions after reform and opening-up policies, while the substantial declines in PM$_{2.5}$ are mainly due to the implementation of the toughest-ever clean air policy in China. In 2020, the nationwide PM$_{2.5}$ concentrations were close to the lowest recorded level in almost 61 years. These two reconstructed PM$_{2.5}$ datasets provide spatiotemporal variations at high resolution, which lay the foundation for research studies associated with air pollution, climate change, and atmospheric chemical reanalysis. It is worth noting that our datasets still have some weaknesses, with the main weakness being a lack of detailed bias estimations for each value in our datasets due to limited historical observations. In the future, we will collect as many PM$_{2.5}$ observations as possible to validate the accuracy of our datasets and provide evaluations of uncertainty for our datasets.

## Acknowledgements

The authors are grateful to all the organization and groups that provided indispensable datasets that we used in this study. We would like to acknowledge to CMA for providing the long-term visibility and MEE for the observational $PM_{2.5}$ data. Also, we acknowledge the MEIC team and Peking for providing the emission inventories and GDEM data provider.

## Financial support

This research was supported by the Major Project from Natural Science Foundation of China (42090030) and the Distinguished Young Scholars Project from Natural Science Foundation of China (41825011).

## Author Contributions

XZ designed the research and led the overall scientific questions. JZ, KG, and LG carried out data processing and analysis based on suggestions from ZZ, DW, YW, and HC. LL calibrated $PM_{2.5}$ data and meteorological observations. YF performed visibility conversions from class values to numeric values before 1980. JL and LJ provided calibrated visibility from 2013 to 2016 and suggestions about manuscript structures. JZ wrote the first draft of the manuscript, and XZ revised the manuscript. All authors read and approved the final version.

## Competing financial interests

The authors declare no competing financial interests.

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
