# Peer review of "Reconstructing 6-hourly PM2.5 datasets from 1960 to"

_Earth System Science Data, 2022_

## Author Response (AR1)

Authors reply to reviewers' comments:

Dear Anonymous Referees,

Thanks for your careful review of the manuscript. We read the reviewers' comments carefully, have considered and responded to all the reviewers' comments, and revised the manuscript accordingly. My detailed responses, including a point-by-point response to the review and a list of all relevant changes, are as follows:

**Reviewer #1: This paper constructed a long-term $PM_{2.5}$ dataset in China based on visibility measurements and meteorological datasets. The quality of the data is well validated, and the trend and spatial variability of $PM_{2.5}$ in China is also examined. This dataset is very useful in studying the long-term changes of particulate matter pollution as well as aerosol radiative effects in China. The paper is also clearly presented and well written. I only have two comments:**

1. In constructing the $PM_{2.5}$ model, the authors used a set of rigorous variable-selection strategy, and selected a set of variables used for prediction. I wonder if the authors could also provide the relative importance of these variables? Is visibility the most important feature, or the emissions?

**Response:** Thanks kindly for your comment. We have added to the supplement Fig. S1 that shows the relative importance of all the features used in our model. As shown in Fig. S1, visibility from the nearest meteorological station is the most important feature, accounting for over 5% of the overall importance. This is consistent with our previous study. Year, elevation, latitude, distance, and the mean visibility of surrounding meteorological stations followed closely. The relative feature importance of them all exceeded 3%. These features indicate temporal and spatial features, elevation, and location contributions. Emission variables, including $NH_3$, $SO_2$, OC, and BC, are also important for our model but not the most important ones. This might be because that part of their influence on $PM_{2.5}$ was replaced by the visibility that is more correlated with $PM_{2.5}$ variations.

[Figure]

Figure S1. Relative importance of all the features used in our model

2. I think that the organization of the results section can be changed a bit for better logical flow. I understand that the authors logic to separate the discussion into temporal and spatial, and also split the validate into these two parts. But typical the readers would expect the results to be validated in full and then analyzed. So it might be better to move all the validation into a separate "Validation" section, and combine all trends and spatial distribution into a "Spatial-temporal variability" section. This is only my thought, the author can decide whether to change."

**Response:** Thanks for your comments. Following the suggestion, we moved the spatial validation section to "3.1 Evaluation of model hindcast performance" (L317-326). In this way, the validation section includes 10-fold CV, by-year CV, spatial CV, and independent validation. This will show the reader all the validation information at once. We also considered putting all the spatial and temporal variations together (i.e., combining sections 3.2 and 3.3), but it will look like too much content in this section after the combination. And we have one site-based PM$_{2.5}$ dataset and one gridded PM$_{2.5}$ dataset, which correspond exactly to section 3.2 and section 3.3, respectively. So we ended up keeping section 3.2 and section 3.3.

**Reviewer #2:** This manuscript reconstructed site-based and gridded PM$_{2.5}$ datasets at six-hour intervals from 1960 to 2020 using visibility, traditional meteorological factors, and other variables based on machine learning methods. These two datasets' quality was well evaluated using 10-fold CV, by-year CV, spatial CV, and independent validation and compared with other available datasets. It shows that the two PM$_{2.5}$ datasets are more advantageous in long-term records and high temporal resolution, which would be of great value for evaluating long-term variations, radiative effects, and health impacts of PM$_{2.5}$ in China. I suggest that this manuscript be published after addressing the following issues:

1. There have been studies on the hourly PM$_5$ estimations based on AOD data from geostationary satellites, such as Himawari 8. However, it needs to be acknowledged that AOD from geostationary satellites is only available during the daytime and the sequence time is relatively short. I suggest adding related studies and pointing out their strengths and weaknesses in the Introduction Section. Also, relationships between PM$_{2.5}$ and visibility together with other meteorological variables have been widely documented in previous studies but lacking in this manuscript, it's better to add relevant studies to make the content of this section more complete.

**Response:** Thanks for your suggestions. We have supplemented relevant studies about estimating hourly PM$_{2.5}$ from geostationary satellites like Himawari 8 (Chen et al., 2019; Yan et al., 2020; Wang et al., 2021; Wei et al., 2021) and pointed out that obtained PM$_{2.5}$ datasets can only extend for several years and the data is missing at night or with cloud cover (L92-95). We also added studies that show the relationship between PM$_{2.5}$ and visibility and conventional meteorological variables (Zhang et al., 2013a; Zhang et al., 2013b; Zhang et al., 2015; Wang et al., 2018; Zhu et al., 2018; Zhong et al., 2018) and specifically introduced the relationship between wind speed, RH and visibility, and PM$_{2.5}$, respectively (L102-110).

2. It is mentioned in the manuscript that extracting spatial features can significantly improve the prediction accuracy of the model, but this is not verified in the manuscript. Adding some sensitivity experiments by setting two groups with/without extracted features will serve to demonstrate their impacts.

**Response:** Thanks for the comments. We set some control groups to show the changes in model performance with/without extracted features in our previous study (Zhong et al., 2021). It's found that without visibility from the nearest station, the R$^2$ value of observed and predicted PM$_{2.5}$ only decreases from 0.75 to 0.72. In contrast, the R$^2$ value decreases more significantly to 0.65 when spatial features of visibility from surrounding stations are excluded. This finding has shown that extracting spatial features can significantly improve the model's prediction accuracy. Therefore, we didn't carry out control experiments in this manuscript to avoid duplication.

3. In Section 3.3., the authors found the large biases among different public available $PM_5$ datasets and proposed to apply ensemble average to multi-datasets. I'm curious about whether the authors consider the specific approach to fusing different $PM_{2.5}$ datasets and how to evaluate the accuracy of the fused dataset.

**Response:** This is an interesting and enlightening question. I think one way to reduce the bias is to average several $PM_{2.5}$ datasets directly, and the other way is to take advantage of $PM_{2.5}$ monitor sites from CMA. CMA has established hundreds of $PM_{2.5}$ sites whose locations do not overlap with MEE sites. The CMA $PM_{2.5}$ dataset from observations can be regarded as an independent validation dataset to evaluate the accuracy of several publicly available $PM_{2.5}$ datasets. This observation dataset can also be regarded as the training target to build a fusing model with publicly available $PM_{2.5}$ datasets as inputs.

4. The authors specify the spatial resolution of the input data for constructing grid points in the text, and the current grid resolution is 0.25°. Is it possible to further improve the resolution while ensuring accuracy?

**Response:** To determine the most appropriate spatial resolution, we need to consider the number distribution density, spacing, and relationship with the spatial resolution of meteorological stations in the target region. In general, the higher the density and the more uniform the distribution within the station, the better the regional meteorological conditions can be reflected, and the spatial resolution can be further improved with a certain precision. Limited by the numbers of the national meteorological stations, we are not sure about the accuracy after continuing to improve the resolution. In future work, we will introduce more regional meteorological stations to increase the spatial resolutions of our dataset and perform a series of control groups with different resolutions to evaluate their performance.

5. What is the duration for the hourly meteorological records mentioned in the manuscript (L139)? Did they start in 1960 or in recent years? Please point it out.

**Response:** From September 2013 to 2016, visibility measurements gradually shifted from 6-hourly manual observations to 1-hourly automatic observations site-by-site. Thus, our meteorological observations include 6-hourly records between 1960-2020, partly hourly records between 2013-2016, and hourly records between 2017-2020. This has been added to the manuscript (L150).

6. Are the CV results in Fig. 2 hourly, 6-hourly, or daily? It's better to point out the time resolution in the title of Fig. 2.

**Response:** The time resolution for CV results is hourly and 6-hourly between 2013-2016 and hourly between 2017-2020. This has been added to the title of Fig. 2 (L280-281).

7. L423: The word "The" in "For by-year CV, The…" should be lowercase.

**Response:** Thanks for your reminder. "The" has been changed to lowercase (L436).

8. L416: The verb be in "The sited-based PM2.5 dataset are in the CSV format, and the gridded dataset PM2.5 are…" should be singular.

**Response:** Thanks for the correction. This has been revised (L429).

Chen, J., Yin, J., Zang, L., Zhang, T., and Zhao, M.: Stacking machine learning model for estimating hourly PM2.5 in China based on Himawari 8 aerosol optical depth data, Science of The Total Environment, 697, 134021, https://doi.org/10.1016/j.scitotenv.2019.134021, 2019.

Wang, B., Yuan, Q., Yang, Q., Zhu, L., Li, T., and Zhang, L.: Estimate hourly PM2.5 concentrations from Himawari-8 TOA reflectance directly using geo-intelligent long short-term memory network, Environmental Pollution, 271, 116327, https://doi.org/10.1016/j.envpol.2020.116327, 2021.

Wang, X., Dickinson, R. E., Su, L., Zhou, C., and Wang, K.: PM 2.5 Pollution in China and How It Has Been Exacerbated by Terrain and Meteorological Conditions, Bulletin of the American Meteorological Society, 99, 105-119, 10.1175/bams-d-16-0301.1, 2018.

Wei, J., Li, Z., Pinker, R. T., Wang, J., Sun, L., Xue, W., Li, R., and Cribb, M.: Himawari-8-derived diurnal variations in ground-level PM2.5 pollution across China using the fast space-time Light Gradient Boosting Machine (LightGBM), Atmos. Chem. Phys., 21, 7863-7880, 10.5194/acp-21-7863-2021, 2021.

Yan, X., Zang, Z., Luo, N., Jiang, Y., and Li, Z.: New interpretable deep learning model to monitor real-time PM2.5 concentrations from satellite data, Environment International, 144, 106060, https://doi.org/10.1016/j.envint.2020.106060, 2020.

Zhang, H. L., Wang, Y. G., Hu, J. L., Ying, Q., and Hu, X. M.: Relationships between meteorological parameters and criteria air pollutants in three megacities in China, Environmental Research, 140, 242-254, 10.1016/j.envres.2015.04.004, 2015.

Zhang, R., Li, Q., and Zhang, R.: Meteorological conditions for the persistent severe fog and haze event over eastern China in January 2013, Science China Earth Sciences, 57, 26-35, 10.1007/s11430-013-4774-3, 2013a.

Zhang, X., Sun, J., Wang, Y., Li, W., Zhang, Q., Wang, W., Quan, J., Cao, G., Wang, J., Yang, Y., and Zhang, Y.: Factors contributing to haze and fog in China, Chinese Science Bulletin, 58, 1178, 10.1360/972013-150, 2013b.

Zhong, J., Zhang, X., Dong, Y., Wang, Y., Liu, C., Wang, J., Zhang, Y., and Che, H.: Feedback effects of boundary-layer meteorological factors on cumulative explosive growth of PM2.5 during winter heavy pollution episodes in Beijing from 2013 to 2016, Atmos. Chem. Phys., 18, 247-258, 10.5194/acp-18-247-2018, 2018.

Zhong, J., Zhang, X., Gui, K., Wang, Y., Che, H., Shen, X., Zhang, L., Zhang, Y., Sun, J., and Zhang, W.: Robust prediction of hourly PM2.5 from meteorological data using LightGBM, National Science Review,

8, 10.1093/nsr/nwaa307, 2021.

Zhu, W., Xu, X., Zheng, J., Yan, P., Wang, Y., and Cai, W.: The characteristics of abnormal wintertime pollution events in the Jing-Jin-Ji region and its relationships with meteorological factors, Science of the Total Environment, 626, 887-898, 2018.